



**Combining METEOSAT-10 satellite image data with GPS tropospheric path delays**
**to estimate regional Integrated Water Vapor (IWV) distribution**
Anton Leontiev[1] and Yuval Reuveni[2,3,4]
[1]*Department of Electrical Engineering, Ariel University, Ariel, Israel*
[2]*Samaria and the Jordan Rift regional R&D Center, Ariel University, Ariel, Israel*
[3]*Department of Mechanical Engineering & Mechatronics, Ariel University, Ariel, Israel*
[4]*School of Sustainability, Interdisciplinary Center (IDC) Herzliya, Herzliya, Israel*





**Abstract:**

Using GPS satellites signals, we can study different processes and coupling mechanisms that can help us understand the physical conditions in the upper atmosphere, which might lead or act as proxies for severe weather events such as extreme storms and flooding. GPS signals received by ground stations are multi-purpose and can also provide estimates of tropospheric zenith delays, which can be converted into mm-accuracy Precipitable Water Vapor (PWV) using collocated pressure and temperature measurements on the ground. Here, we present the use of a dense regional GPS networks for extracting tropospheric zenith path delays combined with near Real Time (RT) METEOSAT-10 Water Vapor (WV) and surface temperature pixel intensity values (7.3 and 12.1 $\mu m$ channels, respectively) in order to obtain absolute IWV (kg/m$^2$) or PWV (mm) distribution. The results show good agreement between the absolute values obtained from our triangulation strategy based solely on GPS Zenith Total Delays (ZTD) and METEOSAT-10 surface temperature data compared with available radiosonde Precipitable IWV/PWV absolute values. The presented strategy can provide unprecedented temporal and special IWV/PWV distribution, which is needed as part of the accurate and comprehensive initial conditions provided by upper-air observation systems at temporal and spatial resolutions consistent with the models assimilating them.




## 1. Introduction:


Water vapor is a greenhouse gas, which can lead to global warming. It repetitively cycles
through evaporation and condensation, transporting heat energy around the Earth and
between the surface and the atmosphere [*Solomon et al.*, 2010]. Water vapor in the
atmosphere contracts the short wavelength radiation of the sun to propagate through the
atmosphere, but traps the long wavelength radiation emitted by the Earth's surface [*van*
*Vleck*, 1947]. This trapped radiation causes temperatures to increase. As the temperatures
increase, the air can sustain a larger amount of water vapor, thus magnifying the
greenhouse effect [*Duan et al.*, 1996]. Since water vapor is the most variable component
of the troposphere, investigation of its distribution and motion is of great importance in
meteorology and climatology [*Soden et al.*, 2004]. Although it plays a key role in
determining climate sensitivity, our current ability to constantly monitor changes in water
vapor at high spatial resolution is insufficient [*Kley et al.*, 2000]. This problem manifest
the most in the upper troposphere making accurate in situ measurement a challenging
task due to the small concentrations of water vapor [*Soden et al.*, 2004].

There are several approaches for estimating the amount of water vapor at the troposphere.
The most common ones utilize radiosondes [*Kley et al.*, 2000; *Soden et al.*, 2004;
*Miloshevich et al.*, 2006], different techniques of the GPS meteorology [*Bevis et al.*,
1992, 1994; *Duan et al.*, 1996; *Ware and Alber*, 1997], or measurements from remote
sensing satellites [*Velden et al.*, 1997; *Jiang et al.*, 2012]. Radiosondes offer an essential
component of the global observing system due to their extended lifetime and broad
geographic coverage [*Kley et al.*, 2000]. Radiosondes have long been the main observing



platform for monitoring tropospheric WV, and are still widely used to provide water
vapor profiles both for field campaigns and as part of national observing networks [*Soden*
*et al.*, 2004]. Radiosondes observations have the advantage of delivering high vertical
resolution, acquisition under clear and cloudy conditions and a long historical record
[*Soden and Lanzante*, 1996]. However, substantial spatial and temporal discontinuities
frequently related to differences in radiosondes instrumentation have also been well
documented [*Elliott and Gaffen*, 1991; *Soden and Lanzante*, 1996; *Free and Durre*,
2002]. Furthermore, there are still national observing networks (i.e., the Israel
Meteorological Service (IMS)), which conduct upper-air measurements to characterized
the temporal behavior of atmospheric boundary layer from a single permanent sounding
site [*Dayan and Rodnizki*, 1999]. This makes it almost impossible to precisely detect the
horizontal boundaries between moist and dry air, especially when most radiosondes are
launched at 12-h intervals and delivers limited temporal resolution [*Moore et al.*, 2015].

When electromagnetic signal travel through the troposphere they are delayed and therefor
slowed down. The amount of delay depends primarily on the pressure, temperature, and
water vapor content, which vary constantly in space and time [*Reuveni et al.*, 2015].
Geophysicists and geodesists have developed methods for estimating the degree to which
signals propagating from GPS satellites to ground-based GPS receivers are delayed by
atmospheric water vapor [*Wdowinski and Eriksson*, 2009]. This delay is parameterized in
terms of a time-varying zenith wet delay (ZWD) that is recovered by stochastic filtering
of the GPS data [*Bevis et al.*, 1992, 1994; *Duan et al.*, 1996].



Satellite observations of the upwelling IR (infrared) radiation in the WV absorption bands
can also provide a unique source of information on tropospheric WV [*Soden and*
*Lanzante*, 1996]. Within the thermal IR domain the European geostationary METEOSAT
satellites are capable of almost continuous monitoring (every 15 minutes using
METEOSAT-10) while observing the earth in the atmospheric window (8.7-13.4 $\mu m$)
and WV absorption frequency band (6.2 and 7.3 $\mu m$). The spatial resolution at the
satellite point corresponds to 5 x 5 km$^2$ for the IR and WV channels. The METEOSAT IR
and WV channel observations are taken in the engineering quantity "count" mode, and
has to be converted into equivalent physical "radiance" unit [*Schmetz et al.*, 1997]. The
calibration is accomplished by linking the observed clear sky WV pixel values to a
calculated radiance at the top of the atmosphere as determined by radiative transfer
calculations using temperature and humidity profiles from radiosondes. This could lead to
bias errors of up to 5%, which corresponds to approximately 1 K in brightness
temperature. The main advantage of using satellite data such as METEOSAT is the
ability to obtain water vapor distribution on regional or global scale [*Roca et al.*, 1997].

We argue that using GPS meteorology coupled with METEOSAT surface temperature
and WV interpolated data can produce adequate results for water vapor estimation. Here,
we present our results for estimating water vapor content around Israel and the Middle
East area using different techniques, comparing their validity and choosing the best
strategy for estimating water vapor distribution.

**2. Technical Approach and Methodology:**



In this paper we calibrate METEOSAT WV pixel values for Israel area using precipitable
water (PW) or integrated water vapor (IWV) obtained from all available GPS stations
around Israel area (Figure 1). First, we estimate PW/IWV values above each GPS station
using the Jet Propulsion Laboratory's (JPL's) GIPSY-OASIS precise point positioning
(PPP) software and tropospheric products [*Zumberge et al.*, 1997; *Bertiger et al.*, 2010;
*Reuveni et al.*, 2012, 2014, 2015]. The PW/IWV estimation is based on tropospheric
ZWD and gradient, tropospheric dry delay, and surface temperature values. Second, WV
pixel values obtained from METEOSAT-10 images are found for the GPS stations
location.  Finally, a mathematical dependency is found between the two data sets which
allow us to transform the entire METEOSAT-10 WV pixel values to absolute WV values
accordingly.

**2.1 PW/IWV estimation from GPS**
The GPS data retrieved from the SOPAC archive (http://sopac.ucsd.edu/) are from
stations of the Survey of Israel (MAPI) GPS network. The GPS data were processed
separately for each day using the Jet Propulsion Laboratory's (JPL's) GIPSY-OASIS
precise point positioning (PPP) software and products. A 7° minimum elevation cut-off
for the satellite observations was applied along with the Vienna Mapping Function 1
(VMF1; *Boehm et al.*, 2006). Zenith hydrostatic delay (ZHD) values from the VMF1
Grid were used every 6 hours. The GIPSY-OASIS software used in this study considers
the tropospheric zenith delay and gradients as stochastic parameter to enable time varying
behavior. Stochastically time varying parameters are assumed to be constant within each
time step, but may change from one time step to another. After a measurement has been



processed (and the parameter estimation had been updated), a time update is executed,
adding process noise to the parameter uncertainties in order to simulate unmodeled or
mismodeled effects [*Reuveni et al.*, 2012]. The tropospheric zenith wet delay and the
gradient parameters are allowed to vary within 5.0e–8 km/√sec (corresponds to about 3
mm in an hour) and 5.0e–9 km/√sec (corresponds to about 0.3 mm in an hour),
respectively. Once the ZWD value is obtained for a specific time interval (i.e. 5 minutes)
the IWV can be easily calculated using the surface temperature [*Bevis et al.*, 1992]:
$$IWV = \kappa ZWD \tag{1}$$
where
$1/\kappa = 10^{-6}(k_3/T_m + k_2/R_v)$, $k_3 = 3.776 \cdot 10^5 K^2 mbar^{-1}$, $k_2 = 64.79 mbar^{-1}$, $R_v$ is
the specific gas constant for water vapor, and $T_m$ is the weighted mean temperature.
Furthermore, $T_m$ might be simply approximated with:
$$T_m = 70.2 + 0.72 T_s \tag{2}$$
where $T_s$ is the surface temperature. For our GPS PW/IWV estimations we used the
nearest surface temperature values measured by the Israel Meteorological Service (IMS)
to each GPS site (http://www.ims.gov.il/IMSEng/All_tahazit/). Figure 2 represent the
PW/IWV values extracts for HRMN GPS station (N33°18'30", E35°47'07") using the
procedure described above. In order to validate that the GPS PW/IWV estimations are
accurate, we compared them to the absolute PW values estimated from IMS radiosonde
data (Figure 3), which is considered as the most accurate method for obtaining PW
measurement. The comparison between the two data sets shows a high correlation
($R^2$=0.97) for all available data during the year 2015 (approximately 240 days).





While processing the entire Israel GPS network data we discovered that precise
temperature measurements for all the GPS station location couldn't be fully obtained due
to the fact that several IMS stations are outside our predefined GPS surrounding area
(>10 km radius). Within a network of 24 permanent GPS stations which are designated to
deliver full spatial coverage for 20,000 km$^2$ area, the surface temperature data for each
GPS location is critical for establishing the mathematical dependency between the GPS
PW and METEOSAT-10 PW data sets. One way to solve this problem is to use the
12 $\mu m$ METEOSAT-10 IR channel to estimate the surface temperature at the GPS station
locations. A comparison between the surface temperature estimation from METEOSAT-
10 and IMS measurements is shown in Figure 4. The correlation between the two is fairly
good (R$^2$=0.79), and usually the difference between the two does not exceed 2°C.
However, temperature differences may be higher when satellite image pixel falls near
water sources (such as the Mediterranean sea, Dead sea, Gulf of Aqaba and lake
Kinneret), and the measured pixel value is averaged between the ground and water
temperatures (Figure 5). Averaging the surrounding pixels values above a pre-determined
threshold can help reducing these temperature differences. For example, we took the
exact pixel, which corresponds to the exact station location and then averaged the square
3x3 pixels around the station. Each pixel of Meteosat-10 image covers area ranging from
3x3 km$^2$ up to 11x11 km$^2$, depending on the longitude and latitude. For Israel area, each
pixel covers an area of approximately 5x5 km$^2$.

In spite of the moderate correlation (R$^2$=0.79) between the surface temperatures obtained
from the 12 $\mu m$ METEOSAT-10 IR channel and other available measured temperature





sources (on-site reading or IMS stations) used for estimating WV throughout GPS
tropospheric path delays, using the METEOSAT-10 surface temperature values produces
approximately similar WV absolute values. Figure 6 represent the comparison between
WV estimation using GPS ZWD along with IMS surface temperatures and GPS ZWD
along with METEOSAT-10 surface temperature. The correlation between the two is very
high ($R^2$=0.99) and indicates that using GPS ZWD along with METEOSAT-10 surface
temperatures for estimating IWV can also reach accurate absolute values. These results
can be simply explained due to the fact that the extracted IWV has a stronger dependency
on GPS ZWD rather than the measured surface temperatures.

## 2.2 WV estimation from METEOSAT-10

Meteosat-10 WV (6.2 and 7.3 $\mu m$) images represent the slant path between the satellite
and a specified point at the Earth's surface (rather then the vertical WV amount above the
point). Therefore, the satellite image pixel values should be normalized at each point to
obtain the vertical path (Figure 7a). Under the assumption that IWV is distributed
uniformly around the Earth, we can obtain a straightforward normalization function
$k(\phi, \lambda)$, which is longitude and latitude dependent:

$$k = (L - l)/h \tag{3}$$

where

$$l = \frac{2(r+H)cos\beta - \left((2*(r+H)cos\beta)^2 - 4(H-h)(H+h+2r)\right)^{1/2}}{2} \tag{4}$$

$$L = ((r + h)^2 + r^2 - 2rHcos\alpha)^{1/2} \tag{5}$$

$$\alpha = acos(cos\phi cos\lambda) \tag{6}$$



$$\beta = sin\left(\frac{r sin\alpha}{L}\right) \tag{7}$$


where $\phi$, $\lambda$ are the latitude and longitude, respectively, H is the height of the
geostationary orbit (H = 35786 km), h is the height of the troposphere (h = 10 km), and r
is the Earth's radius (r = 6371 km). The term in Equation (3) depends strongly on the ratio
between the troposphere height and the distance from the point at the surface to the
satellite. For our present estimations, we assume the troposphere height is equal to 10 km.
The troposphere extends upwards above the boundary layer, and ranges in height from an
average of 9 km at the poles, to 17 km at the Equator. Consequently, for regional areas
this height might be calculated more precisely using regional neutral atmosphere models
or in situ measurements that take into account horizontal inhomogeneities and some other
factors (such as winds, air flows and convection). The dependency of the function given
in Equation (3) on latitude and longitude is shown in Figure 7b.

Once all METEOSAT-10 WV image pixels are normalized, we can extract the
mathematical dependency between the satellite pixel values and absolute IWV values
obtained from the GPS ZTD and surface temperature values. The dependency between
the satellite normalized pixel values and GPS IWV is shown in Figure 8. Using the Least
Squares (LS) method (or any linear fitted polynomial function) we can obtain the relation
between the two:
$$IWV = -0.396 * pix + 156.630 \tag{8}$$
where IWV is the GPS integrated water vapor and pix is the satellite normalized image
pixel value.





## 3. Results


Using the dependency in Equation (8) we can translate the entire image pixel values into
absolute WV values to obtain regional scale distribution (Figure 9). Thus, based on the
dependency of METEOSAT-10 image pixel values on GPS IWV absolute values we are
able to construct regional maps of WV distribution, using only METEOSAT-10 images.
An example for a regional WV distribution map of the surrounding Israel area and
Middle East region, which were produced using the data from METEOSAT-10 7.3 $\mu m$
channel, is shown in Figure 9. The constructed regional maps, with IMS surface
temperatures (a) or METEOSAT-10 12 $\mu m$ IR extracted surface temperatures (b), are in a
good agreement (mean and RMS differences between (a) and (b) are 0.07 and 1.36mm,
respectively).

Although we have shown that it is possible to use the mathematical dependency between
the normalized METEOSAT-10 7.3 $\mu m$ channel and GPS IWV (both with IMS surface
temperatures or METEOSAT-10 12 $\mu m$ IR extracted surface temperatures), the most
straightforward approach for constructing regional WV maps would be interpolating
sufficient GPS IWV data along the desired region. For Israel area, there are currently 24
permanent GPS stations which are fully operational, but the data is not always available.
For example, the largest number of GPS station that we could find (using the SOPAC
archive) during the year 2015 was 16. Still, when all available GPS data is interpolated
using Delaunay triangulation (bilinear interpolation) for each specified date and time, an
accurate (compered with PW radiosondes measurements) regional WV map can be
constructed. Since the interpolation is implemented in a region of highly varying terrain,





it is important to take the topography into account instead of interpolating across terrain
features [*Reuveni et al.*, 2015]. Consequently, the WV estimates at points above sea level
height (h) are scaled to sea level (sl), using a scale height (S) for the wet delays as:
$$N_{sl} = N_h e^{-\frac{h}{S}} \tag{9}$$
The scale heights used for the wet delay is 3000 m [*Means*, 2011; *Means and Cayan*,
2013]. After applying the interpolation at sea level, the interpolated WV field is then
separately scaled to terrain elevation using the identical scale heights and a 6-arcsec
DEM. Figure 10 represents the regional WV map produced from the above specified
triangulation procedure for August 21, 2015 at 12:00.

As mentioned above, the best way to determine the regional WV map (constructed from
triangulating all available GPS data) accuracy is to compare the WV values above the
exact location where the radiosonde measurements are taken (i.e. at Bet Dagan site). For
that matter, we produced 9 consecutive WV maps between March 1 and March 9, 2015,
and compared the values at each map above Bet Dagan site to the absolute radiosonde
WV measurements (Figure 11). The correlation between the two data sets was very high
($R^2$=0.94). Furthermore, the constructed GPS WV regional maps using the triangulation
procedure can be used as a reference grid (for areas inside the maps that are overlapped
since the triangulation can be applied only within the GPS network) for assessing the
construct regional maps of WV distribution extracted from the normalized METEOSAT-
10 7.3 $\mu m$ channel. Comparison between the two technics for August 21, 2015 at 12:00
shows a good agreement with mean and RMS differences of 4.48 and 5.08mm,
respectively (Figure 12). The relatively large differences appear near the mountains (the



Golan Heights and Dead Sea) where the METEOSAT-10 pixel resolution fails to capture
small changes in the topography and presents the averaged WV estimations.

**4. Conclusions**
In this work we have presented 2 different approaches for deriving regional WV
distribution maps; triangulating WV estimations based on GPS ZWD and surface
temperatures extracted from METEOSAT-10 12 $\mu m$ IR channel, or converting
METEOSAT-10 7.3 $\mu m$ WV pixel values using a mathematical dependency to a known
estimated GPS WV values.

The main advantage of using the converted METEOSAT-10 7.3 $\mu m$ WV pixel values is
that we can potentially produce WV distribution maps using the METEOSAT-10 data
and a small number of GPS station data. The main disadvantage of this technique is the
uncertainty regarding all the extremely high (and low) satellite pixel values. Low pixel
value means that amount of water in the surrounding area is very high, and most likely
this is a cloud. Due to the fact that the emitted satellite radiation cannot penetrate beneath
the cloud, the amount of WV might not be fitted while constructing the dependency.
Therefore, It is useful to combine different channels, e.g. VIS and WV or IR and WV
since the cloud temperature is extremely lower than the ground temperature. The most
common way to measure absolute WV values is using radiosondes, but since it allows
estimating WV values only above one corresponding radiosonde point, it is mostly used
for validating the accuracy of the other technics.



The best way for constructing regional WV maps is by interpolating WV estimations
based on GPS ZWD values, since it allows obtaining the most accurate WV values
distribution for relatively large areas. The results obtained from interpolation are in good
agreement with the measured radiosondes data ($R^2$=0.94). The constructed GPS WV
regional maps can also be used as a reference grid for assessing the construct regional
maps of WV distribution which are extracted from the normalized METEOSAT-10
7.3 $\mu m$ channel. Comparison between two techniques shows that the constructed
METEOSAT-10 WV maps fails to take into account small changes in the topography
(i.e. mountains which are consist of both highland and lowland). For example,
differences at the Golan Heights and Dead Sea are extremely large due to the relative
small resolution of METEOSAT-10 sensors (5x5 km$^2$/pixel), which causes METEOSAT-
10 images present the averaged values of WV from the 5x5 km$^2$ square.

Furthermore, we also conclude that the temperature obtained from METEOSAT-10
12 $\mu m$ IR channel can be used for GPS WV precise calculations while using it along with
the ZWD estimations. However, a special care is needs when using the satellite inferred
surface temperature due to the existent of clouds and surrounding water areas.
Comparison of VIS and IR bands might help to exclude clouds and reduce inaccuracies in
while extracting surface temperatures. The presented strategy discussed above can
provide unprecedented temporal and special IWV/PWV distribution, which is needed as
part of the accurate and comprehensive initial conditions provided by upper-air
observation systems at temporal and spatial resolutions consistent with the models
assimilating them.





**Acknowledgments**
Continuous GPS data were provided by SCIGN operated by the Scripps Orbit and
Permanent Array Center (SOPAC). This work was founded by the Israeli Minister of
Science, Technology & Space grant 3-11687.

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



**Figure 1:**


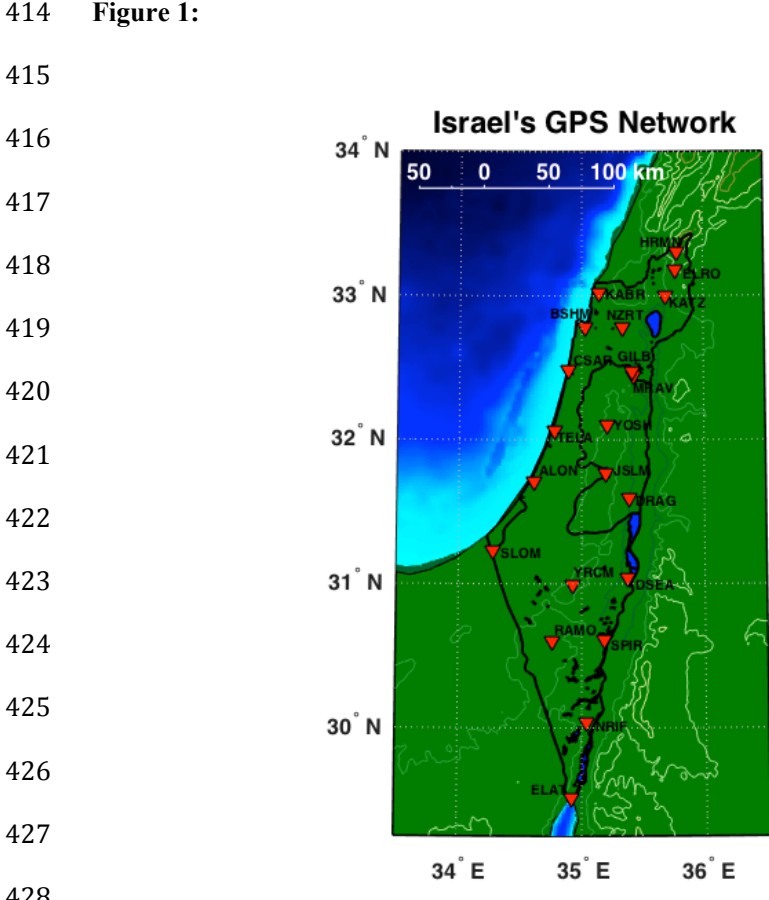


**Figure 1: Israel's GPS network.** The network is maintained by the Survey of Israel
(MAPI) and is consisted of 24 permanent geodetic GPS receivers.








**Figure 2:**

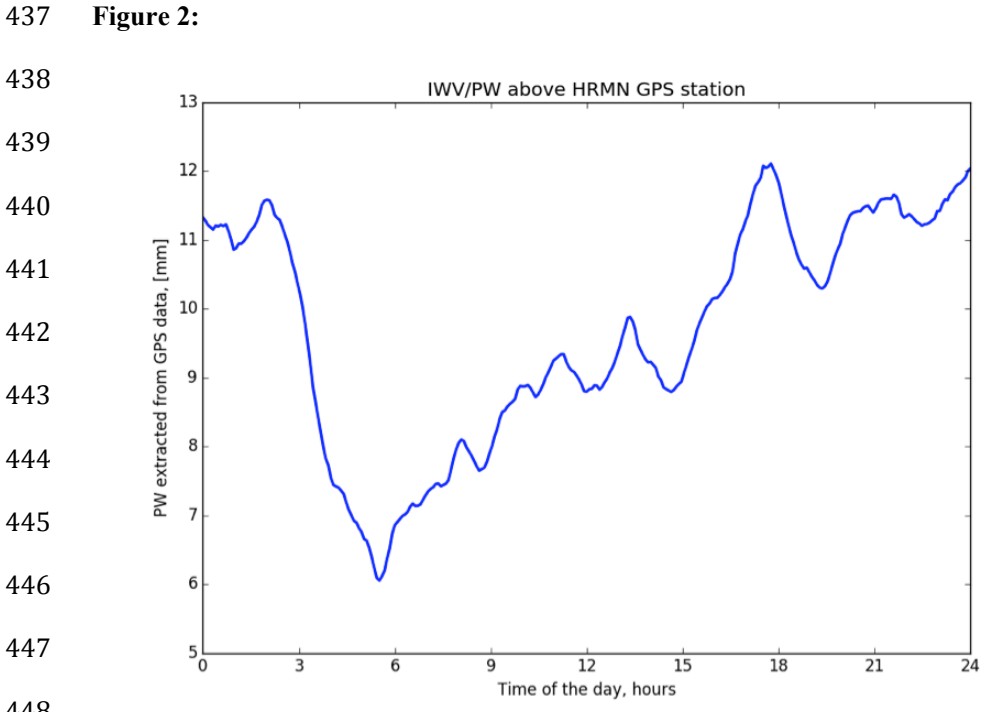

**Figure 2:** Distribution of IWV/PW above HRMN GPS station during May 25, 2015.






**Figure 3:**
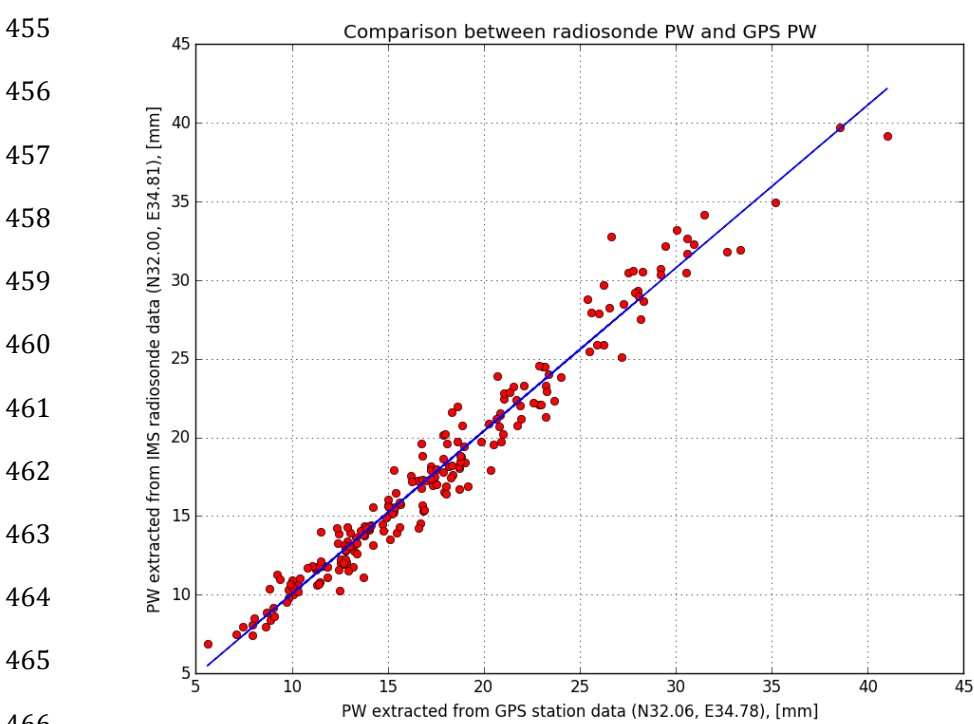

**Figure 3:** Comparison between PW [mm] extracted from IMS radiosonde and GPS data
for approximately 240 days during the year 2015. The correlation shows good agreement
($R^2$=0.97) between the two data sets. GPS PW values were estimated from ZWD and
IMS surface temperature measurements.










**Figure 4:**

Figure 4: Comparison between Meteosat-10 and IMS temperature measurements. The blue line represents a linear fit ($R^2$=0.79) for the temperatures values obtained from Meteosat-10 and IMS.

**Figure 4:** Comparison between Meteosat-10 and IMS temperature measurements. The
blue line represents a linear fit ($R^2$=0.79) for the temperatures values obtained from
Meteosat-10 and IMS.












**Figure 5:**

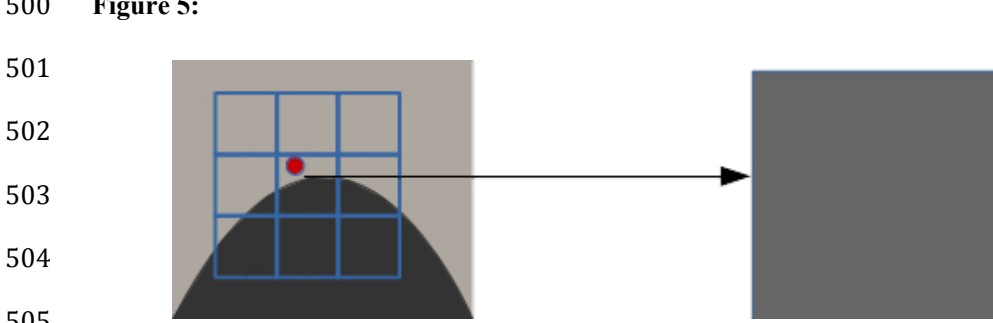

**Figure 5:** Problematic satellite image pixels which fall near water sources, and the actual
measured pixel value is averaged between the ground and water temperatures. Dark areas
represent water source (low temperatures), while the light areas represent the surrounding
ground (high temperatures). The red point represents the location for ground station
surface temperature measurements. The actual averaged pixel value is shown on the right.

















**Figure 6:**

**Figure 6:** Comparison between IWV obtained using GPS ZWD along with Meteosat-10 surface temperatures ($12\,\mu m$ IR channel) and GPS ZWD along with IMS measured temperatures. The correlation between the two is very high ($R^2$=0.99), indicating that the extracted IWV has a stronger dependency on GPS ZWD rather than the measured surface temperatures.









**Figure 7:**

a).






b).


**Figure 7:** a). Conceptual geometry of the satellite slant and vertical paths relative to the
Earth's surface. b). Normalization function (Equation (1)) for latitude and longitude
dependency.





**Figure 8:**

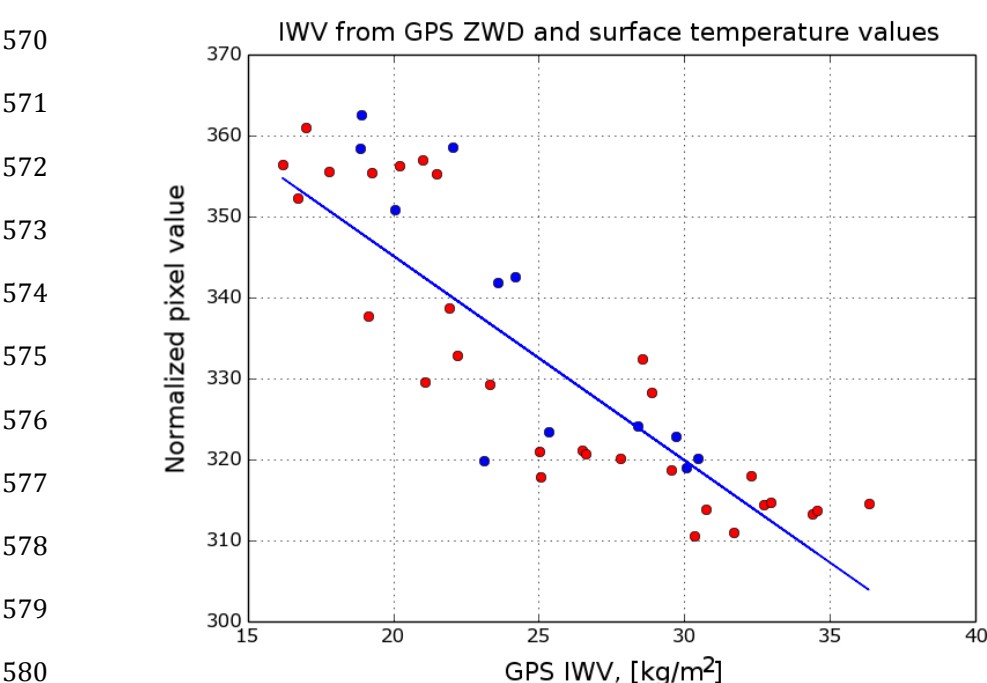


**Figure 8:** Extracting the dependency between METEOSAT-10 normalized pixel values
and GPS IWV absolute values (using surface temperatures from IMS stations). Red
points represent the GPS stations which were taken for extracting the dependency. The
blue line represents dependency, obtained using least Squares (LS) method and. Blue
points represent the GPS stations which were used for validating the extract LS
dependency.





**Figure 9:**
a).

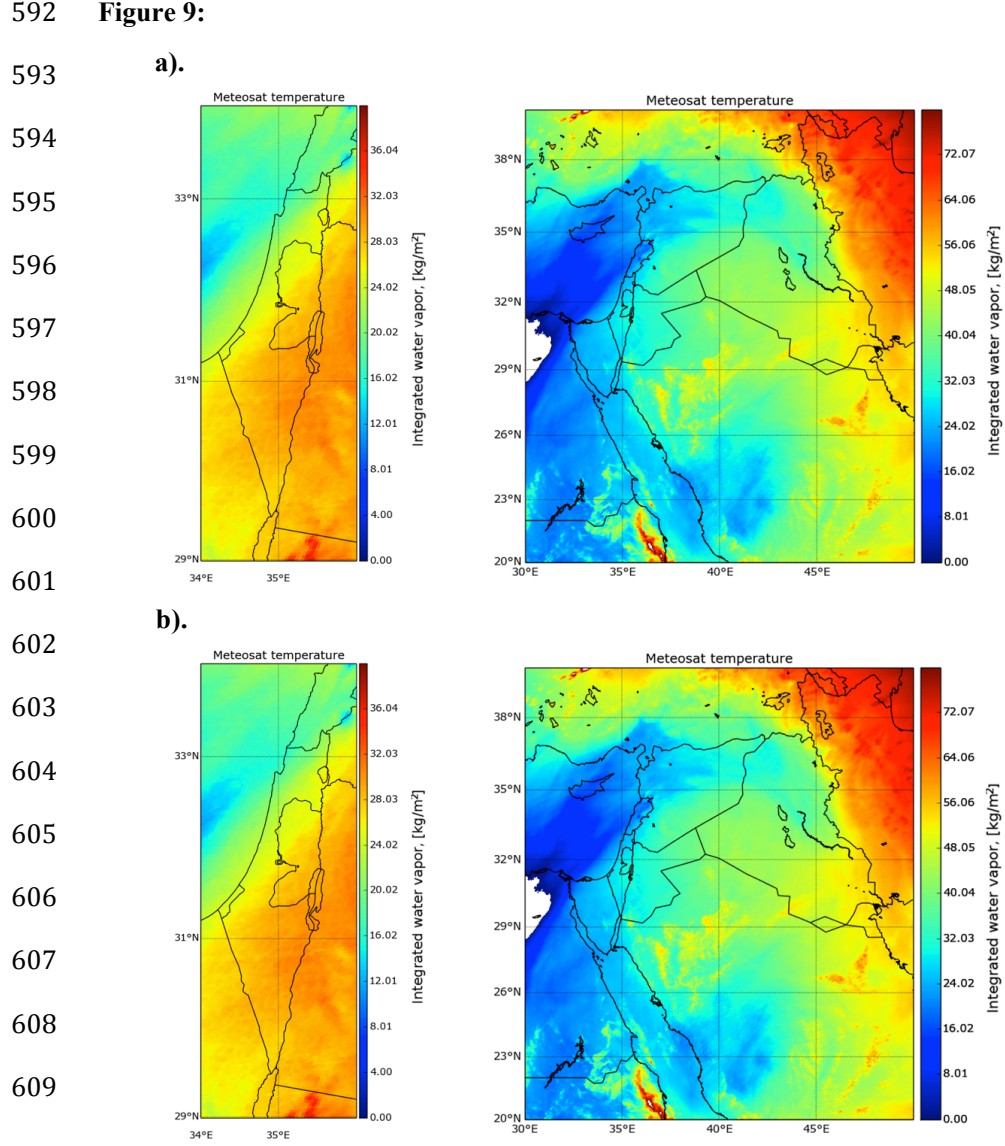

b).

**Figure 9:** Regional WV Distribution maps above Israel area (left) and for the entire
Middle East region (right) constructed from METEOSAT-10 7.3 $\mu m$ channel for August
21, 2015 at 12:00. Necessary surface temperatures were obtained from: (a) IMS stations,
or (b) METEOSAT-10 12 $\mu m$ IR channel. Mean and RMS differences between (a) and
(b) are 0.07 and 1.36mm, respectively.




**Figure 10:**

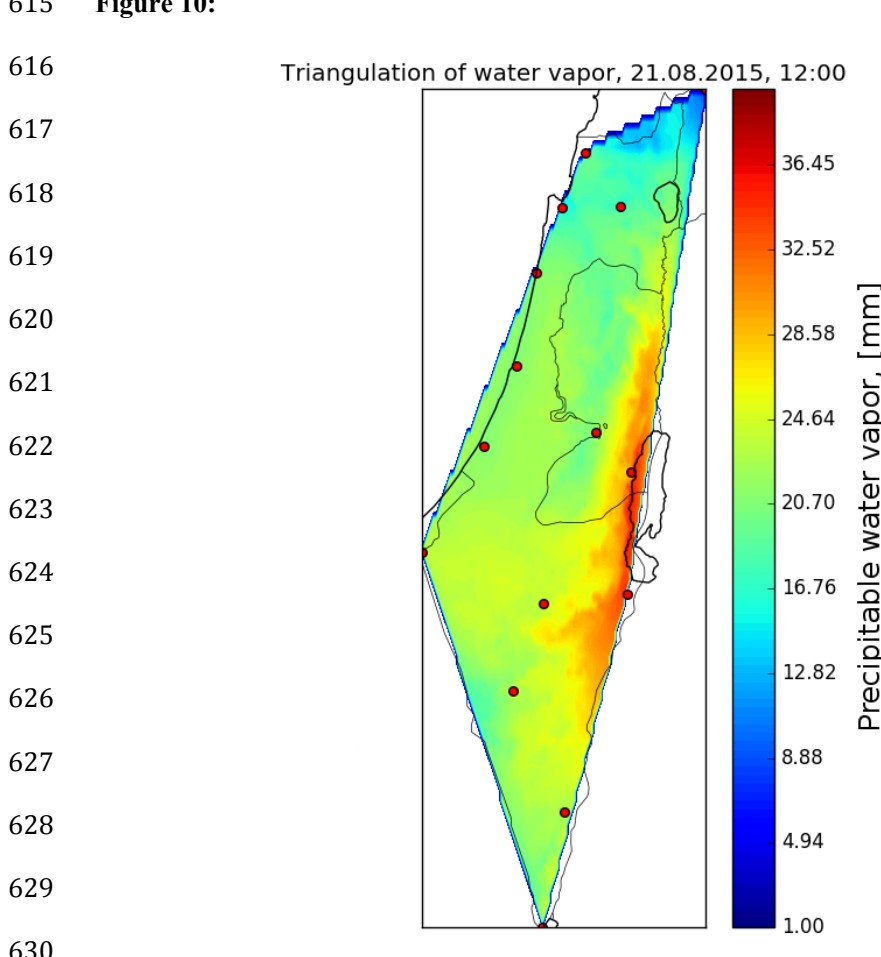

**Figure 10:** Triangulation of PWV above Israel for August 21, 2015. Red dots represent

all available GPS stations (16 in number) that were accounted for.





633    **Figure 11:**

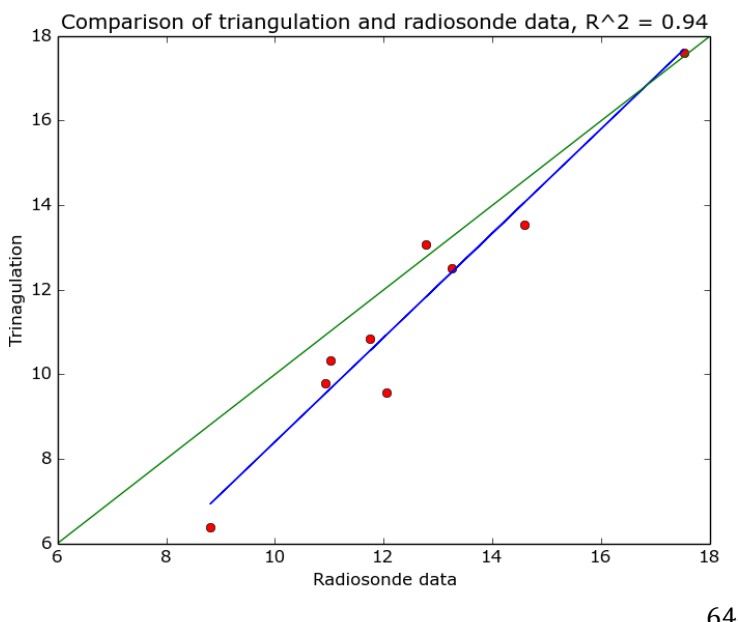


**Figure 11:** Comparison between triangulation procedure and absolute PW value obtained
from radiosondes measurements for 9 consecutive days (between March 1 and March 9,
2015). Red dots represents the data points, blue line represents the Least Square (LS) fit,
green line represents the area where both data sets are completely equal.











**Figure 12:**

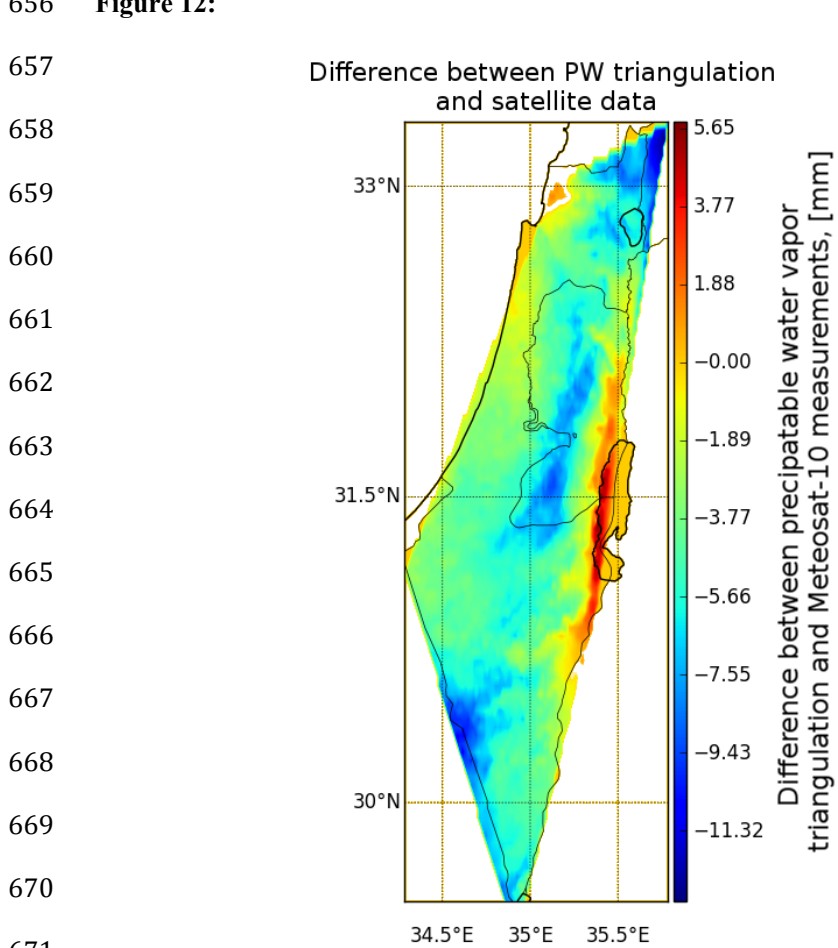

**Figure 12: Comparison between triangulated GPS-WV and WV Distribution maps**
**constructed from METEOSAT-10 7.3 $\mu m$ channel for August 21, 2015 at 12:00.**
Comparison between the two shows a good agreement with mean and RMS differences
of 4.48 and 5.08mm, respectively. METEOSAT-10 pixel resolution fails to capture small
changes in the topography and presents averaged WV estimations above the Golan
Heights and Dead Sea.