# Peer review of "Combining METEOSAT-10 satellite image data with GPS tropospheric path delays"

_Atmospheric Measurement Techniques, 2016_

## Referee Comment (RC1) · Anonymous Referee #2 · 29 Aug 2016

**Evaluation review of the manuscript** amt-2016-217 "Combining METEOSAT-10 satellite image data with GPS tropospheric path delays to estimate regional Integrated Water Vapor (IWV) distribution by A. Leontiev and Y. Reuveni

**General comments:**

This is an interesting study, which provides a description and results of experimental application of a potentially useful methodology for estimation of the regional distributions of integrated water vapour (IWV) based on satellite measurements opening new perspectives for climate and weather prediction research applications. However, to be accepted for publication in Special Issue: Advanced Global Navigation Satellite Systems tropospheric products for monitoring severe weather events and climate (GNSS4SWEC) the manuscript needs a revision including a deeper discussion of results of earlier achievements in the area, presenting additional results of verification vs existing observational data and editing.

**Major comments**

1) Despite the availability of remote sensing measurements contemporary weather and climate analyses still significantly rely on conventional observation data. With the progress in the understanding mechanisms responsible for extreme weather and climate events the need for a more active use of the satellite data becomes clear. The study under the evaluation represents a new step in this direction which is focused on the determination of distribution of vertically integrated water vapor over the area of Israel based on GPS meteorology coupled with METEOSAT surface temperature. Results of application of the methodology in Israel are analyzed based on the mean for the whole period (about 240 days) values of the correlation between IWV and air temperature time series. A more detailed evaluation is performed for only one day (1200 UTC August 21, 2015). Presenting and discussing of additional information on the accuracy of the calculations (e.g. maximum errors, absolute and mean errors) and especially their variation during the year.

2) Only a relatively short discussion of the earlier research efforts is presented. Please consider the possibility of discussing of some of the following publications.

M. P. Cresswell, A. P. Morse , M. C. Thomson,  S. J. Connor (1999) Estimating surface air temperatures, from Meteosat land surface temperatures, using an empirical solar zenith angle model, International Journal of Remote Sensing, 20:6, 1125-1132, DOI: 10.1080/014311699212885

G. Guerova, J. Jones, J. Dousa, G. Dick, S. de Haan, E. Pottiaux, O. Bock, R. Pacione8, G. Elgered, and H.  Vedel  (2016) Advanced global navigation satellite systems tropospheric productions for monitoring severe weather events and climate (GNSS4SWEC) DOI: 10.13140/2.1.3351.2968 ·

S. Hagemann, L. Bengtsson, G. Gent (2003) On the determination of atmospheric water vapour from GPS measurements, J. of Geophys. Res,. 108, NO. D21, 4678, doi:10.1029/2002JD003235

S. Heise, G. Dick, G. Gendt, T. Schmidt, and J. Wickert (2009) Integrated water vapor from IGS ground-based GPS observations: initial results from a global 5-min data set Ann. Geophys., 27, 2851–2859, 2009www.ann-geophys.net/27/2851/2009/

Hordyniec P. , Bosy J, Rohm  W (2015) Assessment of errors in Precipitable Water data derived from Global Navigation Satellite System observations, Journal of Atmospheric and Solar-Terrestrial Physics, 129, Pages 69–77

Vedel, H., Huang, X.-Y., Haase, J., Ge, M., Calais, E. (2004) Impact of GPS Zenith Trophospheric Delay data on precipitation forecasts in Mediterranean France and Spain In: Geophysical Research Letters, 31, No. 2, L02102. DOI: 10.1029/2003GL017715

Wang, J., Zhang, L., and Dai, A.: Global estimates of water-vapor-weighted mean temperature of the atmosphere for GPS applications, J. Geophys. Res., 110, D21101, doi:10.1029/2005JD006215, 2005.

**Editorial comments**

Line 39  "exceptional distribution" – do you mean "resolution"?
Line 78 -79 "conduct upper-air measurements to characterized the temporal behavior of atmospheric boundary layer" - why in the BL only? Please rephrase.
Line 84 "When electromagnetic signal (s?) travel through the troposphere they are delayed and therefor (e?)." - Please correct.
Line 85 "amount of delay" (??) - Please consider rephrasing.
Line 86 "vary constantly" –  significantly?
Line 93 "upwelling IR" - (Upwelling is just an oceanographic phenomenon) – Please rephrase
100 -101 "WV channel observations are taken in the engineering quantity "count" mode, and has (have?) to be converted into equivalent physical "radiance" unit " - (units?), please consider correcting, rephrasing.
Line 106-107"the main advantage to obtain" - ??? please rephrase.
Line 125 "allow us"- (allows?)
132 -132 "A 7° minimum elevation cut-off for the satellite observations was applied along with the Vienna Mapping Function" - Please clarify or rephrase.
134 - 138 "(VMF1; *Boehm et al.*, 2006). Zenith hydrostatic delay (ZHD) values from the … but may change from one time step to another"- This para is not clear. Please rephrase.

Line 171-171 - Your statement "The correlation between the two is fairly good ($R^2$=0.79)" contradicts to line 183 "moderate correlation ($R^2$=0.79) between the surface temperature" – please rephrase.

Line 192 "...can be simply (?) explained due (?) to the fact that the extracted IWV has a stronger dependency" – Please rephrase.

Line 251 "accurate (compered with PW radiosondes measurements)" – compared?

Line 244 245 and line 263 "the most straightforward approach … "the best way"

Line 275 "The relatively large differences appear near" – please provide the value.

Line 293 "Therefore, It (it?) is useful to"

Line 309 "relatively small (???) resolution of METEOSAT-10 sensors (5x5 $km^2$/pixel)" – Do you mean high?

Line 314 "However, a special care is needs" – Please correct (is required?)

Line 315 "surface temperature due to the existent (existence?) of clouds" – please correct.

Line 317 "The presented strategy discussed above (the last two words are probably not necessary?)" – Please check.

Line 318 "provide unprecedented temporal and special IWV/PWV distribution" Why it is unprecedented" – please explain or rephrase..

Line 319 -320 "part of the accurate and comprehensive initial conditions provided by upper-air observation systems at temporal and spatial resolutions consistent with the models assimilating them" -Please rephrase. Please consider using "comprehensive observation data for application in modern data assimilation systems required for increase in the accuracy of the forecasts with the contemporary state of science regional numerical weather prediction systems".

---

## Referee Comment (RC2) · Anonymous Referee #1 · 7 Sep 2016

Use of GPS derived Water Vapour (WV) in Europe is a well established techniques but there exist a large difference on regional level. While west and central Europe the topic has reached maturity in south and particularly east Europe it is currently under development. This paper presents the first results of GPS derived water vapour for Israel. Covering this region is a much needed positive development however this paper has major weaknesses, which make the study incomplete and needs to be addressed in full before proceeding to publication. Below is the summary:

Abstract:

[Figure]

The first sentence "can help us to understand the physical conditions in the upper atmosphere" is incorrect. To the best of my knowledge GPS Meteorology niche is the lower atmosphere. The term "upper atmosphere/troposphere" is incorrectly used in the entire paper and I can advice the authors to seek collaboration with atmospheric scientists to cover the obvious gaps of knowledge in their team.

Introduction:

The paper lack review of the-state-of-the-art in the GPS meteorology and Meteosat methods and water vapour products. The focus in the introduction is the WV derived with radiosonde (RS). Further problem is the use of misleading or general statements. For example:

1) line 59-60 "this problem manifests the most in the upper troposphere" - incorrect

2) line 78-79 ". . ., which conduct upper-air measurements to characterised the temporal behaviour of atmospheric boundary layer" - incorrect and misleading.

3) line 80-81 "This makes it almost impossible to precisely detect the horizontal boundaries between moist and dry air" - what do you mean here?

4) line 86 "vary constantly" - not clear does it "vary" or is "constant".

5) The terms WV/IWV/PWV are mixed in the text and also figures, which makes poor impression and makes the paper difficult to read. Also different units are used "kg/m2" and "mm" through the paper which is not helpful.

In short the introduction is not focused and lacks: 1) review of the previous studies in the GPS Meteorology and products from satellites and 2) clearly defined aim and objectives of the study. Thus it is not acceptable in this form.

Technical Approach and Methodology:

The proposed in this section method to derive WV from Meteosat is not convincing.

2.1 PW/IWV

1) It is not clear how ZWD is obtained and what is its accuracy.

2) The requested surface observation radius is 10 km. It is unclear why such narrow radius is selected and preferred. Published studies suggest that the appropriate radius of surface observations can be much lager. Unless sensitivity studies are done the selection of this radius seems arbitrary.

3) Missing is information of derivation of surface temperature from Meteosat. The accuracy of this products is also not clear.

2.2 WV

1) The proposed WV extraction from Meteosat is not convincing. Without proper treatment of the bottom part of the atmosphere this procedure is incomplete and thus the poor comparison reported in section 3. Methods to derive WV product from Meteosat WV channel have been published in the literature and it is advisable to review those methods or use the processed by Meteosat WV products.

2) In my opinion the proposed Least Squares procedure (equation 8) to link GPS-IWV and Meteosat pixel value is not very appropriate. It will likely smooth the high temporal and spatial variability of WV. Thus it needs to be demonstrated that this procedure is appropriate on day to day basis.

Results:

1) The advantage to use Meteosat temperature to surface observations is not clearly demonstrated. The large difference of "1.36 mm" is likely contributed by the accuracy of Meteosat product.

2) The good agreement between the GPS and RS IWV is poorly demonstrated. Statistic with 9 points is not really meaningful. A proper comparison will need to be done covering preferably one year of observations in all seasons.
3) The reported large mean difference of 4.48 mm between GPS and Meteosat WV likely reflects the proposed method for derivation of Meteosat products in section 2.2.

Conclusions:

Incomplete statements needs to be carefully reviewed and corrected.

---

## Referee Comment (RC3) · Anonymous Referee #3 · 9 Sep 2016

Please, see attached.

Please also note the supplement to this comment:
http://www.atmos-meas-tech-discuss.net/amt-2016-217/amt-2016-217-RC3-supplement.pdf

---

## Author Comment (AC1) · 3 Nov 2016

**Answer to Reviewer #1**

We would like to start by thanking you for all the time and effort which you spent reviewing our paper. All your comments, suggestions, and questions were taken into account and all the necessary corrections were made in the revised manuscript.

**General comments**:

This is an interesting study, which provides a description and results of experimental application of a potentially useful methodology for estimation of the regional distributions of integrated water vapor (IWV) based on satellite measurements opening new perspectives for climate and weather prediction research applications. However, to be accepted for publication in Special Issue: Advanced Global Navigation Satellite Systems tropospheric products for monitoring severe weather events and climate (GNSS4SWEC) the manuscript needs a revision including a deeper discussion of results of earlier achievements in the area, presenting additional results of verification vs existing observational data and editing.

We have revised and edited the entire manuscript and added a deeper discussion of results of earlier achievements in the area. We have also added additional results of verification.

**Major comments**:

1. Despite the availability of remote sensing measurements contemporary weather and climate analyses still significantly rely on conventional observation data. With the progress in the understanding mechanisms responsible for extreme weather and climate events the need for a more active use of the satellite data becomes clear. The study under the evaluation represents a new step in this direction which is focused on the determination of distribution of vertically integrated water vapor over the area of Israel based on GPS meteorology coupled with METEOSAT surface temperature. Results of application of the methodology in Israel are analyzed based on the mean for the whole period (about 240 days) values of the correlation between IWV and air temperature time series. A more detailed evaluation is performed for only one day (1200 UTC August 21, 2015). Presenting and discussing of additional information on the accuracy of the calculations (e.g. maximum errors, absolute and mean errors) and especially their variation during the year.

We have extended our detailed evaluation for more days and added all the additional information regarding the accuracy of the calculations and their variation during the year. All the necessary corrections are implemented in the revised manuscript.

2. Only a relatively short discussion of the earlier research efforts is presented. Please consider the possibility of discussing of some of the following publications.

We have revised the introduction part and included more recent related research efforts.

**Editorial comments**:

Line 39 "exceptional distribution" – do you mean "resolution"? We have changed the sentence to: "The presented strategy can provide high temporal and special IWV/PWV resolution…".

Line 78 -79 "conduct upper-air measurements to characterized the temporal behavior of atmospheric boundary layer" - why in the BL only? Please rephrase. We rephrased the sentence to: "which conduct upper-air measurements to characterized the temporal behavior of the lower atmosphere from a single permanent sounding site…"

Line 84 "When electromagnetic signal (s?) travel through the troposphere they are delayed and therefor (e?)." - Please correct. Corrections were made in the revised manuscript.

Line 85 "amount of delay" (??) - Please consider rephrasing. We choose to leave it as is: "When electromagnetic signals travel through the troposphere they are delayed and therefore slowed down. The amount of delay depends mainly on…".

Line 86 "vary constantly" – significantly? "? We have changed the sentence to: "The amount of delay depends mainly on the pressure, temperature, and water vapor content, which vary significantly both in space and time".

Line 93 "upwelling IR" - (Upwelling is just an oceanographic phenomenon) – Please rephrase. We rephrased the sentence to: "of the reflected IR radiation…".

Line 100-101 "WV channel observations are taken in the engineering quantity "count" mode, and has (have?) to be converted into equivalent physical "radiance" unit " - (units?), please consider correcting, rephrasing. Corrections were made in the revised manuscript.

Line 106-107 "the main advantage to obtain" -??? Please rephrase. Corrections were made in the revised manuscript.

Line 125 "allow us"- (allows?) Corrections were made in the revised manuscript.

Line 132 -132 "A 7° minimum elevation cut-off for the satellite observations was applied along with the Vienna Mapping Function" - Please clarify or rephrase. We are simply describing our GPS analysis strategy. This is informative description based on GIPSY-OASIS software and the possible input parameters for obtaining PPP or tropospheric delays. This is a common description among GPS software's users. We choose to leave it as is.

Line 134-138 "(VMF1; *Boehm et al.*, 2006). Zenith hydrostatic delay (ZHD) values from the … but may change from one time step to another"- This part is not clear. Please rephrase. Again, we are simply describing our GPS analysis strategy. This is informative description based on GIPSY-OASIS software and the possible input parameters for obtaining PPP or tropospheric delays. This is a common description among GPS software's users. We choose to leave it as is.

Line 171-171 - Your statement "The correlation between the two is fairly good (R2=0.79)" contradicts to line 183 "moderate correlation (R2=0.79) between the surface temperature" – please rephrase. Corrections were made in the revised manuscript.

Line 192 "…can be simply (?) explained due (?) to the fact that the extracted IWV has a stronger dependency" – Please rephrase. Corrections were made in the revised manuscript.

Line 251 "accurate (compered with PW radiosondes measurements)" – compared? Corrections were made in the revised manuscript.

Line 244-245 and line 263 "the most straightforward approach … "the best way". Corrections were made in the revised manuscript.

Line 275 "The relatively large differences appear near" – please provide the value. The values were added.

Line 293 "Therefore, It (it?) is useful to". Corrections were made in the revised manuscript.

Line 309 "relatively small (???) resolution of METEOSAT-10 sensors (5x5 km2/pixel)" – Do you mean high? Corrections were made in the revised manuscript.

Line 314 "However, a special care is needs" – Please correct (is required?). Corrections were made in the revised manuscript.

Line 315 "surface temperature due to the existent (existence?) of clouds" – please correct. Corrections were made in the revised manuscript.

Line 317 "The presented strategy discussed above (the last two words are probably not necessary?)" – Please check. Corrections were made in the revised manuscript.

Line 318 "provide unprecedented temporal and special IWV/PWV distribution" Why it is unprecedented" – please explain or rephrase… Corrections were made in the revised manuscript.

Line 319-320 "part of the accurate and comprehensive initial conditions provided by upper-air observation systems at temporal and spatial resolutions consistent with the models assimilating them" -Please rephrase. Please consider using "comprehensive observation data for application in modern data assimilation systems required for increase in the accuracy of the forecasts with the contemporary state of science regional numerical weather prediction systems". We changed the sentence according to your suggestions: "The presented strategy can provide high temporal and special IWV/PWV resolution, which is needed as part of the accurate and comprehensive observation data integrated in modern data assimilation systems, and is required for increasing the accuracy of regional Numerical Weather Prediction (NWP) systems forecast".

---

## Author Comment (AC2) · 3 Nov 2016

**Answer to Reviewer #2**

We would like to start by thanking you for all the time and effort which you spent reviewing our paper. All your comments, suggestions, and questions were taken into account and the necessary corrections were made.

**General comments**:

Use of GPS derived Water Vapour (WV) in Europe is a well established techniques but there exist a large difference on regional level. While west and central Europe the topic has reached maturity in south and particularly east Europe it is currently under development. This paper presents the first results of GPS derived water vapour for Israel. Covering this region is a much needed positive development however this paper has major weaknesses, which make the study incomplete and needs to be addressed in full before proceeding to publication. Below is the summary:

**Abstract**:
The first sentence "can help us to understand the physical conditions in the upper atmosphere" is incorrect. To the best of my knowledge GPS Meteorology niche is the lower atmosphere. The term "upper atmosphere/troposphere" is incorrectly used in the entire paper and I can advice the authors to seek collaboration with atmospheric scientists to cover the obvious gaps of knowledge in their team.

Our intention was to point out that GPS technology, in general, can be used to study both lower (IWV) and upper (TEC) atmospheric conditions, however we agree that in the context of severe weather and flooding events, GPS is used to calculate IWV in the troposphere. We changed the description here to lower atmosphere. The term upper troposphere is used in the introduction section when we indicate that the ability to constantly monitor changes in water vapor, at high spatial resolution, is insufficient especially in the upper troposphere due to the small concentrations of water vapor.

**Introduction**:
The paper lack review of the-state-of-the-art in the GPS meteorology and Meteosat methods and water vapour products. The focus in the introduction is the WV derived with radiosonde (RS).

We have revised the introduction part and included more recent related research efforts.

Further problem is the use of misleading or general statements. For example:
1) line 59-60 "this problem manifests the most in the upper troposphere" – incorrect
Corrections were made in the revised manuscript.

2) line 78-79 ": : :, which conduct upper-air measurements to characterized the temporal behavior of atmospheric boundary layer" - incorrect and misleading. Corrections were made in the revised manuscript.

3) line 80-81 "This makes it almost impossible to precisely detect the horizontal boundaries between moist and dry air" - what do you mean here? It is almost impossible to precisely detect the horizontal boundaries between moist and dry air when using a single permanent sounding site which gives a vertical profile.

4) line 86 "vary constantly" - not clear does it "vary" or is "constant". Corrections were made in the revised manuscript.

5) The terms WV/IWV/PWV are mixed in the text and also figures, which makes poor impression and makes the paper difficult to read. Also different units are used "kg/m2" and "mm" through the paper which is not helpful. Corrections were made in the revised manuscript.

In short the introduction is not focused and lacks: 1) review of the previous studies in the GPS Meteorology and products from satellites and 2) clearly defined aim and objectives of the study. Thus it is not acceptable in this form. We have revised the introduction part and included more recent related research efforts.

**Technical Approach and Methodology**:
The proposed in this section method to derive WV from Meteosat is not convincing.
We have revised the technical approach and methodology part

2.1 PW/IWV
1) It is not clear how ZWD is obtained and what is its accuracy. ZWD can be either obtained from: http://garner.ucsd.edu/pub/solutions/gipsy/trop/ or calculated by GIPSY-OASIS software. We calculated it by ourselves using GIPSY-OASIS software and the input parameters for our strategy are now better described. The mean and rms values were added through the entire revised manuscript.

2) The requested surface observation radius is 10 km. It is unclear why such narrow radius is selected and preferred. Published studies suggest that the appropriate radius of surface observations can be much lager. Unless sensitivity studies are done the selection of this radius seems arbitrary. 10 km is the closest distance from the IMS stations to the GPS stations. Based on [*Bai and Feng*, 2003] and Israel's relative small area, we assumed that 10 km radius is reasonable value.

3) Missing is information of derivation of surface temperature from Meteosat. The accuracy of this products is also not clear. The technique which allows to translate METEOSAT-10 images to absolute temperature is described in documents related to

METEOSAT-10, e.g. in PDF_TEN_05105_MSG_IMG_DATA.pdf [*Muller*, 2010]. Briefly, we obtain pixel luminosity and due to the formulas in the document, mentioned      before, translate it into the temperature.

2.2 WV
1) The proposed WV extraction from Meteosat is not convincing. Without proper treatment of the bottom part of the atmosphere this procedure is incomplete and thus the poor comparison reported in section 3. Methods to derive WV product from Meteosat WV channel have been published in the literature and it is advisable to review those methods or use the processed by Meteosat WV products. We have increased the number of points and obtained better results.

2) In my opinion the proposed Least Squares procedure (equation 8) to link GPS-IWV and Meteosat pixel value is not very appropriate. It will likely smooth the high temporal and spatial variability of WV. Thus it needs to be demonstrated that this procedure is appropriate on day to day basis. This procedure might smooth high spatial variations, but not temporal, because of the fact that for the estimation described in equation 8 we used images which represent different seasons and weather conditions.

**Results**:
1) The advantage to use Meteosat temperature to surface observations is not clearly demonstrated. The large difference of "1.36 mm" is likely contributed by the accuracy of Meteosat product. The main advantage of using METEOSAT temperature is the large number of measurements (every 15 minutes) and high spatial resolution (3-11 km, coverage for METEOSAT pixel). It is also shown that in general temperatures from METEOSAT and IMS ground measurements are equal. For estimation of WV values we can use METEOSAT temperature because of the relatively weak dependency of IWV on temperature relative to ZWD.

2) The good agreement between the GPS and RS IWV is poorly demonstrated. Statistic with 9 points is not really meaningful. A proper comparison will need to be done covering preferably one year of observations in all seasons. We have added more points which represent different seasons and weather conditions

3) The reported large mean difference of 4.48 mm between GPS and Meteosat WV likely reflects the proposed method for derivation of Meteosat products in section 2.2. After your constructive comment, we have revised our technique and obtained smaller mean difference – 2.75 mm. The main problem of interpolation is that it is not sensitive to the areas with bigger amount of water vapor (for example, clouds), which are located between stations. For clear sky conditions and flat areas, in our opinion, the results are good in terms of bias and error values.

**Conclusions**:

Incomplete statements needs to be carefully reviewed and corrected. Corrections were made in the revised manuscript.

---

## Author Comment (AC3) · 3 Nov 2016

**Answer to Reviewer #3**

We would like to start by thanking you for all the time and effort which you spent reviewing our paper. All your comments, suggestions, and questions were taken into account and the necessary corrections were made.

**Brief Summary of the Manuscript**
This manuscript attempts to estimate the IWV around the Israel peninsula by combining ground-based GPS derived IWV with METEOSAT-10 IR surface temperature observations. An empirical relationship between METEOSAT-10 pixels and GPS IWV is derived, in order to exploit the potential of METEOSAT-10 observations to provide a complementary observational data set to ground-based GPS stations. The expected analysis could provide a novel technique to remotely sense from space-based platforms the IWV, not only over land regions but also over oceanic locations.

**Major Comments**:
1) The manuscript lacks motivation/objectivity: The authors need to establish the motivation for and objectives of this study. The use of ground-based GPS receivers to retrieve IWV is a well-established technique, and therefore its application to the Israel peninsula does not provide a significant science contribution. What is unique in this analysis? And why this analysis needs to be done? This is one of the first such studies for Israeli area. The article is not only about investigation of GPS derived WV but it is combined with remote sensing results. Besides the estimation of WV using GPS derived ZWD we also tried to estimate METEOSAT derived WV. This analysis needs to be done because it allows estimating WV using only remote sensing data. This way is fast enough and it doesn't require using different software like GIPSY-OASIS or GAMIT-GLOBK. One more advantage is possibility to apply WV estimation in near real time mode; it is as fast as we can obtain the METEOSAT-10 data.

2) The methodology needs explanation: The scientific merit and novelty of this study is the calibration of the METEOSAT-10 satellite observations to infer IWV, which I find quite interesting because it has never been done before. That said, I would like to see a detailed focus/explanation as of how the calibration happens. We have revised and edited the methodology part adding more detailed expiations regarding the calibration procedure.

3) Line 132: The elevation cut-off angle is routinely set to $10^o$. Why did the authors decide to use a cut-off angle of $7^o$? How would their results change if higher elevation angle were used? Does the elevation angle affect the sampling rate of the ground-based GPS receivers? What is the quality of the surface temperature observations of the METEOSAT-10 satellite?

Elevation cut-off is set to $7^o$ based on suggested recipes for using GIPSY strategy. We refer the reviewer to [*Bar-Sever et al.*, 1998]. Mainly it's due to the fact that the effect

of horizontal gradients diminishes quickly as the elevation angle increase. Therefore to sense the gradients it might be necessary to include low elevation angle observations. At the same time, reducing the elevation angle cutoff too much may result in increased errors from multipath and troposphere mapping function. Elevation angle cutoff of $7^\circ$ is considered as a reasonable compromise. The technique which allows to translate METEOSAT-10 images to absolute temperature is described in documents related to METEOSAT-10, e.g. in PDF_TEN_05105_MSG_IMG_DATA.pdf [*Muller*, 2010]. Briefly, we obtain pixel luminosity and due to the formulas in the document, mentioned before, translate it into the temperature. The comparison of temperature from meteorological stations and METEOSAT is shown at the article but it is also shown that even in the case of big differences in temperatures, we don't get a big influence on the final IWV estimations.

4)  Line 200: The water vapor distribution is quite variable over horizontal scales. Why do the authors assume a uniform distribution of IWV? This seems to be a critical component of the calibration process. How does non-uniformity impact the derivation of Equations (3–8)? What is the sensitivity of equation (9) to the choice of values in equations (3–7)?  We realize that WV is quite variable over horizontal scales, but we assumed that the descending air in the subsidence inversion is rather dry and the absorption of radiation is low and the IWV is distributed uniformly around the Earth only for the purpose of projecting correctly the slant to vertical absorption). In our opinion, it is better than providing no normalization at all. Equation (9) doesn't depend on parameters of previous equations; it is used only for taking earth relief into account. This equation represents the vertical distribution of WV.

5)  Figure 4: The linear regression between the METEOSAT-10 and IMS? The blue line fit does not seem optimal. I notice that the slope of the fit should be smaller with a y-intercept around ~ 297.5 K. We have changed this figure and added more data.

6)  Figure 8: What are the 1-sigma uncertainty errors of the fit? 0.49 kg/m$^2$ and mean value is 0.27 kg/m$^2$.

7)  The statistical sampling is rather small and does not guarantee statistical significance. We added more data and we suppose that now it is enough for statistical significance.

8)  How does METEOSAT-10 IWV look like under cloudy conditions? In comparison with GPS IWV it depends on mutual location of the station and clouds. A more detailed description is explained in the revised manuscript.

9) What does the surface temperature error of $2^\circ$ introduce to IWV? It depends on zenith wet delay and temperature range. It might be innaccuracy from 0.5 to 1.5 kg/m.

**Minor comments**

a) **Line 26:** IWV is mostly due the boundary layer water vapor. It does not tell us anything about dynamical processes in the upper troposphere. I consider revising this statement, or remove it complete it because it appears to be out of context. Corrections were made in the revised manuscript.

b) **Line 319:** The statement about the upper air conditions is out of context. Corrections were made in the revised manuscript.

c) **Line 31:** Should read: "network". Corrections were made in the revised manuscript.

d) **Line 53:** Should read: "temperature". Corrections were made in the revised manuscript.

e) **Line 59:** Should read: " manifests". Corrections were made in the revised manuscript.

f) **Line 78:** Should read: "characterize". Corrections were made in the revised manuscript.

g) **Line 35:** Should read: "…bent…"

h) Mention that radiosondes are limited over land regions. ". Corrections were made in the revised manuscript.

i) Mention that radiosondes are radiation-contaminated in the upper troposphere. Corrections were made in the revised manuscript.

j) **Line 84:** Should read: "signals", "therefore". Corrections were made in the revised manuscript.

k) **Line 85:** Should read: "are slowed down". We choose to leave it as is.

l) **Line 96:** Should read: "continuously". Corrections were made in the revised manuscript.

m) **Line 96-97:** The sentence is incomplete. There is something missing. Please, re-write. Corrections were made in the revised manuscript.

n) **Line 153:** Should read: "represents". Also, define what you mean by "nearest". Corrections were made in the revised manuscript.

o) **Lines 154-156:** Current RO missions do not use closed loop tracking. Please, re-write this section.

p) **Line 187:** Should read "represents". Corrections were made in the revised manuscript.

q) Define all variables: k, L, l, beta, alpha in the equations. Also, consider re-writing equation (9), because the alphas are inter-mixed. We leave the equation as is. We used these parametrs in order to make equation 3 more shorter.

r) **Line 273:** Should read "techniques". Corrections were made in the revised manuscript.

s) **Line 293:** Should read "it" Corrections were made in the revised manuscript.
t) **Line 314:** Should read "needed" Corrections were made in the revised manuscript.

We corrected all these mistakes and inaccuracies.